# Tezepelumab: A Potential New Biological Therapy for Severe Refractory Asthma

**DOI:** 10.3390/ijms22094369

**Published:** 2021-04-22

**Authors:** Corrado Pelaia, Giulia Pelaia, Claudia Crimi, Angelantonio Maglio, Luca Gallelli, Rosa Terracciano, Alessandro Vatrella

**Affiliations:** 1Department of Health Sciences, University “Magna Graecia” of Catanzaro, 88100 Catanzaro, Italy; gallelli@unicz.it; 2Department of Medical and Surgical Sciences, University “Magna Graecia” of Catanzaro, 88100 Catanzaro, Italy; giulia.pelaia@gmail.com; 3Department of Clinical and Experimental Medicine, University of Catania, 95131 Catania, Italy; dott.claudiacrimi@gmail.com; 4Department of Medicine, Surgery, and Dentistry, University of Salerno, 84084 Salerno, Italy; angelantonio.maglio@icloud.com (A.M.); avatrella@unisa.it (A.V.); 5Department of Experimental and Clinical Medicine, University “Magna Graecia” of Catanzaro, 88100 Catanzaro, Italy; terracciano@unicz.it

**Keywords:** asthma, alarmins, TSLP, tezepelumab

## Abstract

Thymic stromal lymphopoietin (TSLP) is an innate cytokine, belonging to the group of alarmins, which plays a key pathogenic role in asthma by acting as an upstream activator of cellular and molecular pathways leading to type 2 (T2-high) airway inflammation. Released from airway epithelial cells upon tissue damage induced by several noxious agents including allergens, viruses, bacteria, and airborne pollutants, TSLP activates dendritic cells and group 2 innate lymphoid cells involved in the pathobiology of T2-high asthma. Tezepelumab is a fully human monoclonal antibody that binds to TSLP, thereby preventing its interaction with the TSLP receptor complex. Preliminary results of randomized clinical trials suggest that tezepelumab is characterized by a good safety and efficacy profile in patients with severe, uncontrolled asthma.

## 1. Introduction

Asthma is a chronic obstructive respiratory disease, mainly characterized by airflow limitation due to bronchial inflammation and airway remodeling [1,2]. A hallmark of asthma is the heterogeneity of airway inflammation, expressed by several phenotypes sustained by underlying different endotypes, which consist of complex cellular and molecular pathogenic mechanisms (Figure 1) [3,4,5]. The most frequent endotypes are grouped under the umbrella term “type 2” (T2) asthma, which includes allergic and non-allergic traits, mostly outlined by eosinophilic inflammation [6,7]. Differently from type 2 airway inflammation, T2-low asthma can be featured by either neutrophilic or paucigranulocytic patterns [8,9,10].

In addition to chronic inflammation, all asthma endotypes are often characterized by airway structural changes, which span throughout the various layers of the bronchial wall [11]. These remodeling features include goblet cell metaplasia/hyperplasia, subepithelial fibrosis sustained by activation of fibroblasts and myofibroblasts, smooth muscle thickening, and neo-angiogenesis [11].

In T2-high asthma, the development, persistence, and amplification of eosinophilic inflammation are driven and orchestrated by multiple cellular elements including dendritic cells, T helper 2 (Th2) lymphocytes, group 2 innate lymphoid cells (ILC2), mast cells, basophils, and airway epithelial cells [12,13,14]. Within this endotypic context, a key pathophysiologic role is played by thymic stromal lymphopoietin (TSLP), an innate cytokine especially involved in type 2 eosinophilic inflammation, but also implicated in neutrophilic and paucigranulocytic asthma [15,16,17]. Indeed, TSLP stimulates dendritic cells to guide the differentiation of naïve Th cells towards the Th2 lineage, but can also promote Th17 commitment [15]. Moreover, TSLP activates ILC2, mast cells, and basophils, induces eosinophil survival and transmigration, and also affects the functions of airway structural cells such as fibroblasts and airway smooth muscle cells [15].

Most patients with mild or moderate asthma are well controlled by inhaled corticosteroids (ICS), eventually integrated by the addition of long-acting β_2_-adrenergic agonists (LABA) within ICS/LABA fixed combinations [18]. Furthermore, asthmatics with more severe disease may need additional medications such as leukotriene modifiers, tiotropium, and even oral corticosteroids (OCS) [19,20]. Despite all these treatments, severe asthma can remain uncontrolled, thus requiring adjunctive biological therapies based on the use of monoclonal antibodies directed against immunoglobulins E (IgE), interleukin 5 (IL-5), IL-5 receptor, or interleukin-4 (IL-4) receptor [21,22,23,24,25,26]. These are excellent add-on treatments, but they could not be effective for all patients with severe T2-high asthma, and they do not provide any benefit for T2-low asthmatic patients. In particular, the most relevant goals of biological therapies for severe asthma include the decreases in both exacerbation rate and OCS intake, as recently highlighted by the PONENTE trial (Study NCT03557307, https://clinicaltrials.gov/ website. Poster presentation at the American Academy of Allergy Asthma & Immunology, 26 February–1 March 2021).

In addition to the already approved, above-mentioned anti-asthma biologics, other monoclonal antibodies are currently undergoing clinical investigation, which might potentially guarantee a wider coverage of therapeutic advantages [27]. Among the latter, one of the most promising biologic drugs for asthma treatment is tezepelumab, a fully human monoclonal antibody that specifically interacts with TSLP, thus preventing its binding to the TSLP receptor complex [28]. Given the relevant importance of TSLP as a master player of asthma pathobiology, this alarmin, acting as an upstream inducer of strategic proinflammatory and remodeling pathways, appears to be a potential suitable target for perspective biological therapies of severe asthma. Therefore, tezepelumab deserves close attention as a possible future anti-asthma biologic.

In light of the above considerations, this article aims to review the role of TSLP in asthma pathophysiology, as well as to discuss the therapeutic properties of tezepelumab as an eventual add-on treatment option for severe asthma.

## 2. Pathogenic Role of TSLP in Asthma

Originally identified in thymic stromal cells, TSLP is an innate pleiotropic cytokine belonging to the four-helix-bundle cytokine family and is distantly related to interleukin-7 (IL-7) [29]. Two TSLP variants exist, including a long isoform (159 amino acids) and a short one (60 amino acids), whose expressions are respectively regulated by different gene promoters, which are responsive to distinct patterns of environmental agents [30,31]. Short TSLP is constitutively present in many tissues where it plays a homeostatic role, whilst the production of the long isoform can be induced by proinflammatory stimuli and is increased in asthmatic patients [31]. The long variant of TSLP exerts its biological functions by selectively binding to its cognate receptor (TSLPR), and this interaction is promoted by the electrostatic attraction occurring between the positive charges of TSLP surface and the negative charges of TSLPR [32,33]. The resulting TSLP/TSLPR binary molecular aggregate in turn recruits the α subunit of the IL-7 receptor (IL-7Rα). As a consequence, the assembly of the extracellular ternary complex TSLP/TSLPR/IL-7Rα leads to activation of an intracellular signaling network including Janus kinases 1 and 2 (JAK1/2), signal transducers and activators of transcription 3 and 5 (STAT3/5), as well as nuclear factor κB (NF-κB), mitogen-activated protein kinases (MAPK), and phosphoinositide 3 kinase (PI3K) (Figure 2) [31,32,33]. These signaling pathways lead to activation of genes encoding Th2 cytokines such as IL-4, IL-5, IL-9, and IL-13.

TSLP belongs to the family of alarmins. Indeed, this innate cytokine is mostly released by airway epithelial cells as an alarm signal when tissue injury is triggered by several environmental insults such as allergens, cigarette smoke, airborne pollutants, viruses, bacteria, and chemical and physical irritants [29,34]. Genomic studies have shown that some single-nucleotide polymorphisms (SNPs) of the TSLP gene are associated with the risk of developing asthma [35]. Indeed, in asthmatic patients TSLP expression is increased in both outer and inner surfaces of bronchial epithelial biopsies, as well as in serum, induced sputum, bronchoalveolar lavage fluid (BALF), and exhaled breath condensate [15,36,37,38]. Moreover, airway expression levels of TSLP are correlated with asthma severity and airflow limitation [15]. TSLP mRNA expression is also up-regulated in nasal polyps from patients with aspirin-exacerbated respiratory disease (AERD) [39].

Once secreted from damaged bronchial epithelium, TSLP affects the functions of several immune/inflammatory and structural cells, which co-express TSLPR and IL-7Rα. In particular, ILC2 is included among the most relevant cellular targets of TSLP. Together with other alarmins such as interleukin-25 (IL-25) and interleukin-33 (IL-33), TSLP prolongs ILC2 survival and stimulates these cells to produce large quantities of interleukin-5 (IL-5), interleukin-9 (IL-9), and interleukin-13 (IL-13) (Figure 1) [13,15,40,41]. IL-5 is the most powerful inducer of eosinophilic inflammation, while IL-9 is a growth factor for mast cells, and IL-13 is involved in mucus hypersecretion, bronchial hyperresponsiveness, and airway remodeling [13,14]. In comparison to mild asthmatics, higher numbers of ILC2-expressing IL-5 and IL-13 have been detected in patients with severe asthma and persistent airway eosinophilia, despite high-dosage OCS therapy [42]. Indeed, TSLP has been shown to cause steroid resistance of BALF ILC2, whose cytokine production was not inhibited by dexamethasone in asthmatic patients with elevated TSLP levels [43]. Furthermore, in bronchial biopsy samples taken from subjects with severe asthma, ILC2 co-localized with TSLP-immunoreactive areas [44]. In this regard, it is also noteworthy that a positive correlation has been detected between the ILC2 number and TSLP levels in nasal biopsies from patients with severe asthma and chronic rhinosinusitis [45].

In addition to ILC2, within the context of innate immunity, TSLP also regulates the functions of both eosinophils and eosinophil progenitors, which play a key role in type 2 airway inflammation. In particular, both TSLPR and IL-7Rα are expressed by human eosinophils susceptible to the activating and anti-apoptotic effects of TSLP, which enhances eosinophil survival and potentiates the release of eosinophil-derived chemokines and cytotoxic proteins [46,47]. These biological actions of TSLP are mediated by signaling pathways based on activation of MAPK and NF-κB [46,47]. The relevant pathogenic role of TSLP in eosinophilic inflammation is further corroborated by the correlation between TSLP immunoreactivity and airway eosinophilia, detected in bronchial biopsies from asthmatic patients 24 hours after allergen challenge [48]. Moreover, TSLP promotes the assembly of eosinophilic extracellular traps, including mitochondrial DNA and eosinophilic cationic protein (ECP), which are implicated in innate immune responses against infectious pathogens responsible for tissue injuries at the level of asthmatic airways [49]. In addition to acting on mature eosinophils, TSLP also affects the functions of eosinophil progenitors. In these cells, TSLP increases the expression of the α subunit of the IL-5 receptor (IL-5Rα) and cooperates with IL-5 in inducing eosinophilopoiesis [50]. In addition to contributing to eosinophil differentiation, TSLP, together with IL-33, is also able to stimulate the migration of eosinophil progenitors by up-regulating their production of chemokines such as CCL1, CCL22, and CXCL8 [15,51].

Other cellular targets of TSLP are basophils, whose TSLPR expression can be stimulated by allergens via IgE-dependent mechanisms [15,52]. TSLP promotes basophil differentiation and enhances the expression of basophil activation markers such as CD203c, as well as increases histamine secretion, cytokine synthesis, and eotaxin-dependent basophil migration [15,52]. In patients with allergic asthma, TSLP also up-regulates basophil expression of the ST2 receptor of IL-33 and IL-17RB receptor of IL-25, thus incrementing the susceptibility of basophils to other alarmins [53]. Further inflammatory cells expressing TSLPR include mast cells, which respond to TSLP by increasing their production of type 2 cytokines such as IL-5 and IL-13 [54]. In association with IL-33, TSLP also triggers mast cell generation of prostaglandin D_2_ (PGD_2_), a pleiotropic eicosanoid that exerts a wide spectrum of pathogenic actions in type 2 asthma [39,55]. Moreover, in addition to airway epithelial cells, airway smooth muscle cells adjacent to mast cells (as well as mast cells themselves) release TSLP thereby implementing paracrine and autocrine loops implicated in asthma pathobiology (Figure 1) [56,57].

In regard to the crosstalk between innate and adaptive immunity underlying the cellular pathophysiology of asthma, TSLP exerts a pivotal function by acting at the level of dendritic cells. Indeed, human myeloid dendritic cells express TSLPR, whose activation by airway epithelium-derived TSLP leads to up-regulation of major histocompatibility complex class II (MHC-II) and several co-stimulatory molecules such as CD40 and CD86 [58]. Moreover, TSLP stimulates dendritic cells to secrete CCL17 and CCL22 chemokines [15], which bind to CCR4 receptors located on Th2 cells thereby guiding their transfer from thoracic lymph nodes to the airways [59]. TSLP can also be produced by dendritic cells themselves, which may further amplify type 2 inflammation [60]. A crucial role is played by TSLP in driving dendritic cell-induced polarization of Th2 lymphocytes [61]. In particular, upon TSLP-mediated induction of OX40 ligand (OX40L) expression, dendritic cells promote the differentiation of naïve CD4^+^ T lymphocytes into mature Th2 cells [62]. Therefore, OX40L up-regulated by TSLP acts as a powerful polarizing signal involved in stimulation of T cell commitment towards the Th2 immunophenotype [63]. When purified from patients with allergic asthma, myeloid dendritic cells can be primed to induce Th2 cell expansion by a combination of TSLP and dermatophagoides pteronyssinus-derived allergens [64]. Interestingly, exosomes derived from TSLP-stimulated dendritic cells activate important intercellular communications via up-regulation of OX40L, leading to proliferation of CD4^+^ T lymphocytes, IL-4 biosynthesis, and Th2 cell differentiation [65]. In allergic asthmatics, increased TSLPR expression by dendritic cells can also be linked to OX40L-independent induction of a Th9 cell-mediated immune response [64]. A further contribution to the pathobiology of T2-high asthma is provided by other cellular elements releasing IL-4 such as T follicular helper cells (Tfh), whose differentiation in lung-draining lymph nodes is dependent on OX40L-positive dendritic cells activated by TSLP [66]. Independently of dendritic cells, TSLP can induce Th2 lymphocyte differentiation through the involvement of other intermediary cells, such as CD11c^+^ monocytes/interstitial macrophages [67]. Although T cells are mostly subject to the indirect effects of TSLP mediated by dendritic cells and monocytes/macrophages, this alarmin can also exert direct actions on both CD4^+^ and CD8^+^ T lymphocytes. Indeed, TSLP can directly induce the commitment of naïve CD4^+^ T cells towards the Th2 lineage, as well as the expansion of CD8^+^ T lymphocytes [68,69]. In patients with allergic asthma, induction of type 2 airway inflammation mediated by TSLP is further potentiated by its negative impact on the immunomodulatory action of lung T regulatory (Treg) lymphocytes [70]. In particular, TSLP inhibits the production of the anti-inflammatory cytokine interleukin-10 (IL-10) by Treg lymphocytes, thus impairing the suppressive effects exerted by these cells on allergic proinflammatory pathways [70].

In addition to being implicated in the pathobiology of type 2 airway inflammation, TSLP appears to be involved in T2-low neutrophilic asthma, characterized by a pivotal pathogenic role played by Th17 lymphocytes [71]. In this regard, it is noteworthy that TSLP can induce dendritic cells to release interleukin-6 (IL-6) and interleukin-23 (IL-23), two cytokines which crucially contribute to the differentiation of naïve CD4^+^ T cells into Th17 lymphocytes [72,73]. It has also been shown that TSLP is able to elicit a dual Th cell commitment, leading to the concomitant expansion of both Th2 and Th17 lymphocytes producing IL-4 and IL-17A, respectively [73].

In addition to significantly affecting the functions of immune/inflammatory cells, TSLP extends its biologic actions to airway structural cells. In particular, airway smooth muscle cells express TSLPR and respond to TSLP by increasing their production of IL-6 and IL-8 [74,75]. Airway smooth muscle cells themselves can synthesize and release TSLP upon stimulation mediated by activated mast cells (Figure 1), as well as by interleukin-1β (IL-1β) and tumor necrosis factor-α (TNF-α) [15,57]. Furthermore, bronchial fibroblasts produce TSLP and express TSLPR, so that when these cells are stimulated by TSLP, they increment the production of collagen, α smooth muscle actin, arginase 1, and transforming growth factor β1 (TGF-β1) [76,77]. Hence, such findings suggest that in asthmatic airways TSLP plays a pivotal role in promoting structural changes via a crosstalk occurring between inflammatory and resident cells, the latter including epithelial cells, fibroblasts, and smooth muscle cells.

## 3. Tezepelumab: An Emerging Biologic Therapy for Treatment of Severe Asthma

Tezepelumab is a fully human monoclonal IgG2λ antibody that specifically ligates TSLP at the level of its binding site for TSLPR, thereby impeding the human TSLP–TSLPR interaction [78]. Initially tested in patients with mild allergic asthma, tezepelumab was shown to be capable of inhibiting several effects induced by an allergen challenge [79]. In particular, 31 enrolled patients were randomly assigned to receive either a placebo or three-monthly injections of tezepelumab, administered via the intravenous route at the dosage of 700 mg. When compared to placebo, tezepelumab prevented allergen-dependent bronchoconstriction, as documented on days 42 and 84 by 34.0% and 45.9% improvements of forced expiratory volume in 1 second (FEV_1_), respectively, with respect to allergen-induced FEV_1_ decreases detected on the same days [79]. Tezepelumab also significantly increased the provocative concentration of methacholine causing a 20% FEV_1_ decrement (methacholine PC_20_). Moreover, significant decreases in blood/sputum eosinophils, as well as in fractional exhaled nitric oxide (FeNO) levels, were detected in patients treated with tezepelumab [78]. This drug did not elicit any change in total serum levels of IgE. No serious adverse events were reported during this study [79].

Subsequently, the randomized, double-blind, placebo-controlled, multicenter PATHWAY phase 2b trial was carried out between December 2013 and March 2017 at 108 sites throughout 12 countries [80]. Enrolled participants were current non-smokers aged from 18 to 75 years, who presented with uncontrolled asthma despite treatment with medium-to-high ICS dosages consisting of 250–500 or more than 500 μg/day of fluticasone propionate dry powder inhaler or equivalent, combined with a LABA bronchodilator. Furthermore, recruitable patients were also characterized by a history of either at least two asthma exacerbations or at least one severe asthma exacerbation leading to hospitalization, occurring during the 12 months preceding enrolment. Asthma exacerbations represented an acute or sub-acute worsening of symptoms and lung function with respect to a patient’s usual status [19]. An exacerbation not requiring hospitalization was defined as a worsening of asthma symptoms needing further treatment for at least 3 days with systemic corticosteroids, regardless of the necessity of an emergency room visit. Uncontrolled asthma was also documented by a score of at least 1.5 of the 6-item Asthma Control Questionnaire (ACQ-6), referring to the screening period. Percentages of the predicted normal pre-bronchodilator FEV_1_ values ranged from 40% to 80%, and a positive response to the reversibility test was shown by a post-bronchodilator FEV_1_ increase of at least 12% and 200 mL with respect to baseline measures. Subjects with clinically relevant respiratory diseases other than asthma were excluded from the study. Among 918 screened patients, 584 were randomly subdivided into four groups including a placebo arm of 148 subjects, and three subgroups of patients receiving subcutaneous injections of tezepelumab every 4 weeks, at dosages of 70 mg (low dose: 145 patients), 210 mg (medium dose: 145 patients), or 280 mg (high dose: 146 patients).

The primary goal of the study was to assess at week 52 the effects of tezepelumab on the annualized asthma exacerbation rate (AAER). In comparison to placebo, tezepelumab significantly (*p* < 0.001) decreased AAER by 61%, 71%, and 66% at the dosages of 70, 210, and 280 mg, respectively [80]. This very relevant impact of tezepelumab on AAER was found to be independent of baseline blood eosinophil counts [80]. Moreover, a recent post hoc analysis of the PATHWAY trial showed that at the dosage of 210 mg, tezepelumab decreased AAER to a similar extent in severe uncontrolled asthmatic patients, with or without nasal polyposis [81]. Another post-hoc analysis also demonstrated that tezepelumab lowered asthma exacerbations across all four seasons of the year [82]. Tezepelumab also prolonged the time to the first asthma exacerbation. Furthermore, when considering the secondary outcomes, after 52 weeks of treatment tezepelumab significantly improved the ACQ-6 score in all three interventional subgroups. Tezepelumab also incremented pre-bronchodilator FEV_1_ by 120, 110, and 150 mL in the low-dose, medium-dose, and high-dose groups, respectively [80]. In addition, in all tezepelumab subgroups this biologic drug significantly and persistently down-regulated key biomarkers of type 2 asthma such as blood eosinophil numbers, FeNO levels, and total serum IgE concentrations [80]. However, the remarkable preventive action exerted by tezepelumab on asthma exacerbations occurred regardless of baseline levels of blood eosinophils or other indicators of T2-high inflammation [80], and this result can undoubtedly represent a relevant advantage for tezepelumab with respect to most of the currently approved anti-asthma biologics. Tezepelumab also decreased blood levels of IL-5, IL-13, periostin, and thymus and activation-regulated chemokine (TARC) [83].

With regard to safety and tolerability, the overall occurrence of adverse events, mainly including nasopharyngitis, bronchitis, and headache, was similar across the four study groups [80]. Indeed, 62.2% of the patients assigned to the placebo arm, as well as 66.2%, 64.8%, and 61.6% belonging to the low-dose, medium-dose and high-dose subgroups experienced at least one adverse event, respectively. Because of the occurrence of adverse events, the trial was discontinued by one patient treated with placebo, as well as by two and three recipients of the medium and high doses of tezepelumab, respectively. Similar rates of skin reactions at the level of the injection site were reported by patients undergoing treatment with either the placebo or tezepelumab. No anaphylactic reactions were reported. Anti-drug antibodies were detected in 8.8% of patients belonging to the placebo arm, as well as in 4.9%, 0.7% and 2.1% of patients receiving low, medium, and high doses of tezepelumab, respectively. No neutralizing antibodies were found.

Ongoing phase 2 and 3 studies, aiming to evaluate the efficacy and safety of tezepelumab, include the NAVIGATOR, SOURCE, DESTINATION, and CASCADE trials [83].

NAVIGATOR is a multicenter, placebo-controlled, double-blind and randomized trial, recruiting more than 1000 adults (18–80 years old) and adolescents (12–17 years) with severe asthma not adequately controlled by medium-to-high dosages of ICS, associated with at least another controller drug [83]. The study protocol is based on a 5/6-week screening phase, followed by a 52-week treatment and a 12-week follow-up period. The trial population should include nearly equal percentages of patients with higher and lower than 300/μL blood eosinophil counts. Once again, the primary endpoint is prevention of AAER [84]. Secondary outcomes include the therapeutic effects of tezepelumab on asthma control, health-related quality of life, and lung function [84]. Preliminary results suggest that tezepelumab, administered subcutaneously at the dosage of 210 mg every 4 weeks, was able to achieve the primary goal of lowering AAER at week 52 [83]. This effect was detected across all enrolled patients, and even in those having less than 300 and 150 blood eosinophils/μL.

SOURCE is another multicenter, 48-week, double-blind, randomized and placebo-controlled trial, enrolling 150 severe asthmatic patients on treatment with medium-to-high doses of ICS/LABA combinations, associated with an additional chronic OCS therapy [85]. The primary aim of this study is to evaluate the eventual OCS-sparing action of tezepelumab, injected subcutaneously at the dosage of 210 mg every 4 weeks [85]. The objective of this study is very important, given the widespread use of OCS among patients with severe asthma, who are thus exposed to the risk of developing the relevant systemic adverse effects of these drugs (infections, hypertension, diabetes, gastrointestinal complications, osteoporosis, bone fracture, glaucoma, cataract, reduced growth in children and adolescents, adrenal insufficiency, and psychiatric disorders) [86].

Outside of the phase 3 PATHFINDER program, which includes both NAVIGATOR and SOURCE studies, DESTINATION is a further phase 3, 1-year trial recruiting 960 patients who have already completed one of the above two studies [87]. Therefore, DESTINATION is a long-term extension study primarily aimed to evaluate the safety profile of tezepelumab in subjects with severe asthma, throughout 104 weeks of therapy inclusive of the previous treatment period (either the NAVIGATOR or SOURCE trials) [87]. A secondary goal of DESTINATION is to investigate the long-term impact of tezepelumab on AAER.

Finally, CASCADE is a phase 2, double-blind, randomized and placebo-controlled study, focusing on the anti-inflammatory activity of tezepelumab [88]. Here, 116 patients with uncontrolled, moderate-to-severe asthma, undergoing treatment with 210 mg of tezepelumab administered subcutaneously every 4 weeks for 28 weeks, will be monitored throughout this study period [88]. Asthmatic subjects will be enrolled regardless of their baseline eosinophilic inflammatory pattern. With respect to baseline, the CASCADE trial aims to evaluate the potential inflammatory changes induced by the biological therapy with tezepelumab. In particular, bronchoscopic biopsies will be examined in order to assess the eventual differences occurring between baseline and week 28 with regard to airway submucosal infiltration by eosinophils, neutrophils, T lymphocytes, and mast cells [88]. Moreover, the possible anti-remodeling action of tezepelumab will be explored by measuring the reticular basement membrane thickening before and after treatment with tezepelumab.

## 4. Conclusions

TSLP represents one of the most suitable therapeutic targets for emerging biological treatments of severe asthma [89,90,91]. Indeed, TSLP plays a strategic role in asthma pathobiology, due to its top position located at a very upstream level within the complex network of cellular and molecular pathways leading to airway inflammation (Figure 1). A blockade of upstream mediators can thus have a more extensive therapeutic impact than antagonism of downstream cytokines and receptors, thus driving the pathogenic mechanisms underlying restricted phenotypes/endotypes. In this regard, the fully human anti-TSLP monoclonal antibody tezepelumab is, so far, the most studied drug among the currently developed anti-alarmins [90]. The initial results provided by the PATHWAY trial open very promising perspectives for an eventual key role of tezepelumab in the future therapies of type 2 severe asthma. Currently, ongoing studies are essential to eventually confirm and extend such preliminary observations. It should be very interesting to comparatively evaluate the therapeutic effects of both currently available and prospective biological treatments of severe T2-high asthma. However, published studies refer to systematic reviews and network meta-analyses [92] rather than to direct head-to-head comparisons, which might prove to be more useful. Furthermore, the potential efficacy of tezepelumab should also be explored with regard to its possible therapeutic application in T2-low neutrophilic asthma, given the relevant contribution of TSLP to the pathophysiology of this difficult-to-treat endotype. Lastly, because of the biologic activities exerted by TSLP on airway structural cells, current trials will probably also provide interesting information about the supposed ability of tezepelumab to modulate bronchial remodeling in severe asthma.

## Figures and Tables

**Figure 1 ijms-22-04369-f001:**
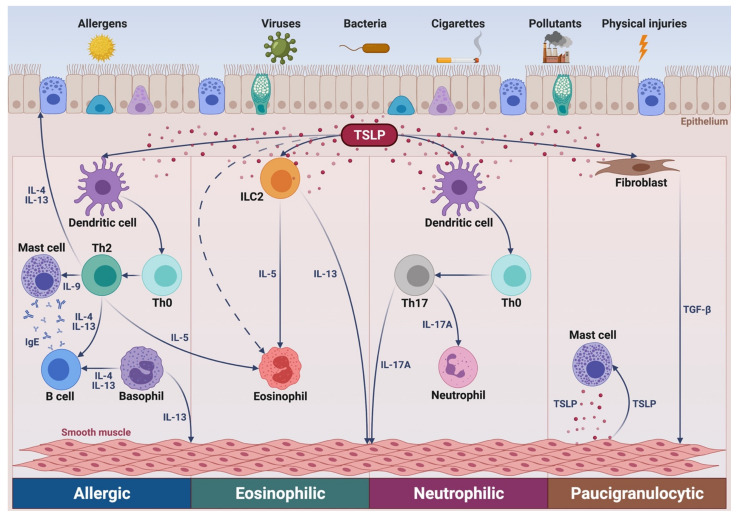
Putative role of TSLP in several asthma pathways. In allergic asthma, via activation of dendritic cells, TSLP promotes the differentiation of Th2 lymphocytes secreting IL-4, IL-5, IL-9, and IL-13, which target B cells, eosinophils, mast cells, and airway smooth muscle cells, respectively. IL-4 and IL-13 are also produced by basophils. In non-allergic eosinophilic asthma, TSLP stimulates ILC2 to release IL-5 and IL-13. In neutrophilic asthma, TSLP induces dendritic cells to drive the development of neutrophil-activating Th17 lymphocytes. In paucigranulocytic asthma, TSLP mediates the complex crosstalks involving inflammatory cellular elements, such as mast cells, and airway structural cells including epithelial cells, fibroblasts, and smooth muscle cells. TSLP: thymic stromal lymphopoetin; Th: T helper; ILC2: group 2 innate lymphoid cells; IL: interleukin; TGF-β: transforming growth factor-β. This original figure was created by the authors using BioRender.com.

**Figure 2 ijms-22-04369-f002:**
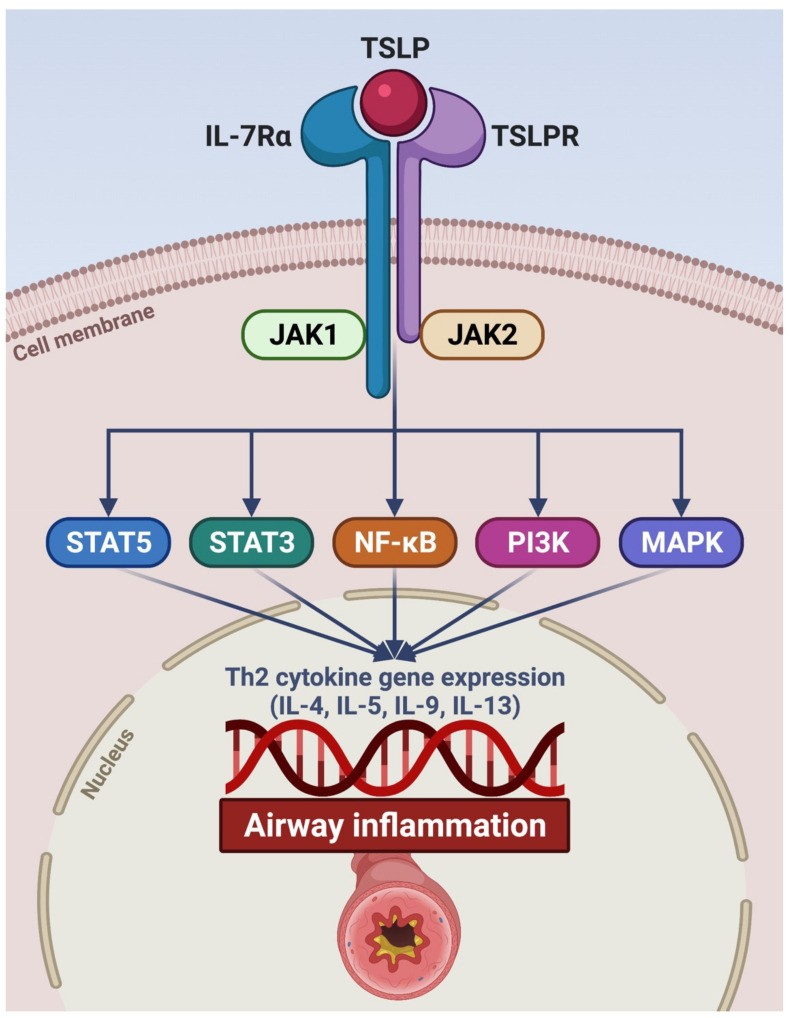
Signaling pathways activated by TSLP. TSLP specifically binds to TSLPR expressed by target cells. The TSLP/TSLPR binary aggregate thus recruits IL-7Rα, thereby promoting the assembly of the ternary molecular complex TSLP/TSLPR/IL-7Rα. The latter triggers the activation of intracellular signaling cascades initiated by JAK1/JAK2, which phosphorylates STAT3/5 and also involves NF-κB, PI3K, and MAPK. As a consequence, this transduction network activates target genes encoding pro-inflammatory cytokines (IL-4, IL-5, IL-9, and IL-13) implicated in airway inflammation. TSLP: thymic stromal lymphopoietin; TSLPR: thymic stromal lymphopoietin receptor; IL-7Rα: α subunit of the interleukin-7 receptor; JAK: Janus kinases; STAT: signal transducers and activators of transcription; NF-κB: nuclear factor-κB; PI3K: phosphoinositide 3 kinase; MAPK: mitogen-activated protein kinases. This original figure was created by the authors using BioRender.com.

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
