# Peer review of "Tezepelumab: A Potential New Biological Therapy for Severe Refractory Asthma"

_ijms, 2021, doi:10.3390/ijms22094369_

Round 1

Reviewer 1 Report

This is a very nicely written review on a hot topic in the field of severe asthma and targeted therapies, where the Authors focus on TSLP and its neutralizing therapy tezepelumab. It is undoubtfully a good timing to report on such review as phase 3 trials are now delivering their findings and it is expected that this therapy will reach the clinics soon. I have only few remarks to further improve the manuscript.

MAJOR COMMENTS.

  1. Introduction: The authors should highlight the need for additional biologics, besides anti-IgE, anti-IL-5(R) and anti-IL4R, namely that those therapies might not capture all T2 or non-T2 patients. For this, mentioning the response rate and the level of ambition which may vary according to the efficacy (cfr e.g. Ponente trial results of benralizumab presented at AAAAI 2021). Seondly, a paragraph on airway fibrosis (relevance and determinants in asthma) could be added in order to set the basis for understanding the related TSLP activity on fibroblasts.
  2. Pathogenic role of TSLP: the first figure is very nice. I would suggest to entitle it "Potential (or putative) role of TSLP in several asthma pathways (and not endotypes) as the described pathways may be overlapping (e.g. ILC2 in both allergic and eosinophilic asthma; fibroblasts and SMC may be important regardless of the immune phenotype). I would also suggest adding an arrow from IL-4/IL-13 to the epithelium (mucus, FeNO) and - dotted- from TSLP to the eosinophil (direct action). In figure 2, the main (inflammatory) genes that are downstream of TSLP and signaling pathways should be mentioned along "airway inflammation". It should also be quoted that TSLPR expression by DCs is increased in allergic asthma (ref 63), and that this feature conditions not only Th2 differentiation but also Th9 differentiation, the later occuring independently from OX40L.
  3. Tezepelumab: this part is really excellent. Maybe a missing paragraph concerns some information on comparative efficacy (indirectly; e.g. see related Reviews and Lebecque P et al. BMJ 2009) and safety (e.g. see Pilette C et al. JACI 2007 and more recent reviews) with approved biologics. The need for head-to-head comparisons could also be mentionned. In addition, the fact that eosinophils are predictive of response for most of the currently approved biologics but not tezepelumab should be highlighted.

MINOR COMMENTS

Typo: "Review" not "elucidate" line 75; IL-7R (and not IL-17R) line 137; delete "however" line 194 and "decline" line 232; replace "complainted of" by "presented with" line 242.

Author Response

Reviewer 1

This is a very nicely written review on a hot topic in the field of severe asthma and targeted therapies, where the Authors focus on TSLP and its neutralizing therapy tezepelumab. It is undoubtfully a good timing to report on such review as phase 3 trials are now delivering their findings and it is expected that this therapy will reach the clinics soon. I have only few remarks to further improve the manuscript.

MAJOR COMMENTS.

  1. Introduction: The authors should highlight the need for additional biologics, besides anti-IgE, anti-IL-5(R) and anti-IL4R, namely that those therapies might not capture all T2 or non-T2 patients. For this, mentioning the response rate and the level of ambition which may vary according to the efficacy (cfr e.g. Ponente trial results of benralizumab presented at AAAAI 2021). Secondly, a paragraph on airway fibrosis (relevance and determinants in asthma) could be added in order to set the basis for understanding the related TSLP activity on fibroblasts.

Answers

The first consideration has been included in the revised text, and the Ponente trial has been cited (pages 2-3, lines 70-79).

Moreover, a short paragraph on airway remodeling has been added in the revised text (page 1, lines 34-38).

  1. Pathogenic role of TSLP: the first figure is very nice. I would suggest to entitle it "Potential (or putative) role of TSLP in several asthma pathways (and not endotypes) as the described pathways may be overlapping (e.g. ILC2 in both allergic and eosinophilic asthma; fibroblasts and SMC may be important regardless of the immune phenotype). I would also suggest adding an arrow from IL-4/IL-13 to the epithelium (mucus, FeNO) and - dotted- from TSLP to the eosinophil (direct action). In figure 2, the main (inflammatory) genes that are downstream of TSLP and signaling pathways should be mentioned along "airway inflammation". It should also be quoted that TSLPR expression by DCs is increased in allergic asthma (ref 63), and that this feature conditions not only Th2 differentiation but also Th9 differentiation, the later occuring independently from OX40L.

Answers

The title of figure 1 has been changed according to the above suggestion, and also the two suggested arrows have been added (page 2).

With regard to figure 2, the main inflammatory genes located downstream of TSLP and signaling pathways have been mentioned in the revised manuscript (page 4).

Furthermore, in the revised text we added the concept referring to TSLP-dependent and OX40L-independent induction of dendritic cell-driven differentiation of Th9 cells (page 6, lines 201-203).

  1. Tezepelumab: this part is really excellent. Maybe a missing paragraph concerns some information on comparative efficacy (indirectly; e.g. see related Reviews and Lebecque P et al. BMJ 2009) and safety (e.g. see Pilette C et al. JACI 2007 and more recent reviews) with approved biologics. The need for head-to-head comparisons could also be mentioned. In addition, the fact that eosinophils are predictive of response for most of the currently approved biologics but not tezepelumab should be highlighted.

Answers

With regard to the comparative efficacy and the need for head-to-head comparisons, an additional paragraph has been included in the revised manuscript (page 9, lines 371-375).

Moreover, the revised text highlights the fact that the therapeutic effects of tezepelumab are independent of baseline blood eosinophil counts (page 8, lines 294-298).

MINOR COMMENTS

Typo: "Review" not "elucidate" line 75; IL-7R (and not IL-17R) line 137; delete "however" line 194 and "decline" line 232; replace "complained of" by "presented with" line 242.

Answers

All the above minor corrections have been made in the revised text.

Reviewer 2 Report

The manuscript is well written and structured narrative review of the effects of TSLP in different types of asthma, and its blocking antibody, tezepelumab. Figures are interesting and appropriate. The bibliography is adequate. 

Just a minor comment. I would suggest including the definition of asthma exacerbation in the clinical trials, as it could help readers. 

By the time of publication, the Navigator trial results will be known. Indeed, some posters have been published. 

Author Response

Reviewer 2

The manuscript is well written and structured narrative review of the effects of TSLP in different types of asthma, and its blocking antibody, tezepelumab. Figures are interesting and appropriate. The bibliography is adequate. 

Just a minor comment. I would suggest including the definition of asthma exacerbation in the clinical trials, as it could help readers.

Answer

The definition of asthma exacerbation has been included in the revised text (page 7, lines 263-265).